# Linking Lichen Metabolites to Genes: Emerging Concepts and Lessons from Molecular Biology and Metagenomics

**DOI:** 10.3390/jof9020160

**Published:** 2023-01-25

**Authors:** Garima Singh

**Affiliations:** Department of Biology, University of Padova, 35122 Padova, Italy

**Keywords:** biosynthetic gene clusters, secondary metabolites, atranorin, grayanic acid, gyrophoric acid, lecanoric acid, physodic/olivetoric acid, usnic acid, lichen metabolites

## Abstract

Lichen secondary metabolites have tremendous pharmaceutical and industrial potential. Although more than 1000 metabolites have been reported from lichens, less than 10 have been linked to the genes coding them. The current biosynthetic research focuses strongly on linking molecules to genes as this is fundamental to adapting the molecule for industrial application. Metagenomic-based gene discovery, which bypasses the challenges associated with culturing an organism, is a promising way forward to link secondary metabolites to genes in non-model, difficult-to-culture organisms. This approach is based on the amalgamation of the knowledge of the evolutionary relationships of the biosynthetic genes, the structure of the target molecule, and the biosynthetic machinery required for its synthesis. So far, metagenomic-based gene discovery is the predominant approach by which lichen metabolites have been linked to their genes. Although the structures of most of the lichen secondary metabolites are well-documented, a comprehensive review of the metabolites linked to their genes, strategies implemented to establish this link, and crucial takeaways from these studies is not available. In this review, I address the following knowledge gaps and, additionally, provide critical insights into the results of these studies, elaborating on the direct and serendipitous lessons that we have learned from them.

## 1. Introduction

Linking molecules to genes is fundamental to customizing products for medicinal use, i.e., to achieve the desired effect and yield, as well as to reduce the potential side effects [1,2,3,4]. Lichens—miniature ecosystems of fungi, cohabitating with one or more photosynthetic organisms and bacteria—are purported to be a treasure chest of bioactive secondary metabolites [5,6,7,8]. For instance, more than one thousand secondary metabolites with great structural and functional diversity have been reported from lichens [7], and many of these have a broad range of therapeutic properties, including antimicrobial, anti-inflammatory, cytotoxic, and antioxidant abilities [5,7,9,10,11]. Despite this richness, the challenges associated with linking lichen metabolites to genes have hindered the exploitation of this vast natural reservoir of natural products (NPs) for commercial production. The surge in genomic resources in the last decade, along with the development of algorithms to identify, compare, and group biosynthetic genes, have greatly assisted in establishing genome-guided links between lichen metabolites and their corresponding genes [12,13,14,15]. Although only a few lichen metabolites have been linked to genes, these studies have led to several interesting breakthroughs, not only related to the genetics of secondary metabolites but also regarding the mechanism and regulation of secondary metabolite synthesis in fungi.

In this review, I highlight the recent advances in linking lichen secondary metabolites to genes and examine the direct and serendipitous discoveries of these studies. I provide insights into how the metagenomics-guided linking of molecules to genes has expanded our understanding of secondary metabolite synthesis and the functioning of biosynthetic genes. This knowledge will lay the foundation for designing future strategies for the genome mining of other lichen metabolites’ genes and will provide a comprehensive framework for cautiously interpreting the results of metagenomics-guided gene discovery. Furthermore, it will build a groundwork for the combinatorial synthesis and genetic engineering of lichen metabolites by shedding light on how the biosynthetic potential of an organism can be fully utilized to generate several metabolites.

This review consists of three parts. In the first part, I elaborate on lichen metabolic potential, the genetics of secondary metabolism, and the non-reducing polyketide synthase (NRPKS), followed by a brief comparison of biosynthetic research on non-lichenized and lichenized fungi. In the second part, I give an overview of the lichen metabolites that have been connected to genes, and provide critical insights into the results of these studies, elaborating on the direct and serendipitous lessons that we have learned from them. In the third part, I summarize the novel concepts of secondary metabolite synthesis and biosynthetic gene function that have emerged from these studies. As the majority of the secondary metabolites produced by lichens are of fungal origin [5,7,8], in this review, the term “lichen metabolites” will be used to refer to the secondary metabolites synthesized by the fungal partner.

## 2. Part I: The Genetics of Secondary Metabolism and Biosynthetic Research on Non-Lichenized Versus Lichenized Fungi

### 2.1. Genetics of Secondary Metabolism 

The genes encoding secondary metabolites are called biosynthetic genes [8,16,17]. The genes involved in the synthesis of a secondary metabolite are usually present in a contiguous fashion as a biosynthetic gene cluster (BGC) [18,19]. A cluster may span up to several kilobases (kb) and contain up to 20 genes. Usually, at least some of the genes present in a BGC are co-regulated and co-expressed [9,20,21]. A cluster is known by the compound that is coded, e.g., aflatoxin BGC, sterigmatocystin BGC, orcinol BGC, etc. Typically, a molecule is coded by a core gene and a few accessory genes (Figure 1A). Apart from these genes, a cluster may also contain one or more regulatory and transport-related genes. The core gene is involved in the synthesis of the backbone of the molecule, which is then modified by accessory (or tailoring) enzymes, such as methylases, oxidases etc., to produce the final molecule [8,16]. The core genes define the chemical class of the molecules. For instance, a PKS cluster has a polyketide synthase (PKSs) as the core gene and produces polyketides from acyl-CoAs; a nonribosomal peptide cluster contains a non-ribosomal peptide synthetase (NRPS) gene, coding for a non-ribosomal peptide from amino acids as starter units; a terpene cluster has terpene synthases and terpene cyclases, which code for the enzymes involved in the synthesis of terpenes from activated isoprene units [8,16,19]. The accessory genes code either for the enzymes involved in the modification of the molecule coded by the core genes, or in the transcription regulation or transportation of the final molecule. Accessory genes can vastly alter the structure and, hence, the bioactivity of the metabolites. Often a BGC codes for several metabolites, depending upon which sets of genes are activated.

### 2.2. PKSs—Structural Diversity

Fungal PKSs are multidomain enzymes. Depending upon the domain present, they can be divided into three major classes, namely, the non-reducing (NR), the highly-reducing (HR), and the partially-reducing (PR) PKSs (Figure 1B). The three classes of PKSs—highly, partially, and non-reducing PKSs—also form well-supported clades, as suggested by the phylogenetic tree inferred from the KS and AT domains [22]. It is suggested that NRPKS evolved from reducing PKSs due to the loss of the reducing domains [22]. 

As for the domains, NRPKSs have a starter ACP transacylase (SAT), ketoacyl synthase (KS), an acyl transferase (AT), one or more acyl carrier proteins (ACP or PP), a product template (PT), and thioesterase (TE) or the releasing domain (Figure 1B) [23,24,25,26]. AT, KS, and ACP are the minimal domains required for polyketide synthesis, on which the main chemical reaction of polyketide synthesis takes place (Figure 1B) [27]. The reducing PKSs have minimal domains plus the reducing domains, such as ketoreductase (KR), dehydratase (DH), and enoyl reductase (ER), which catalyze the reduction of a keto group to a hydroxyl group, dehydration of the hydroxyl to an enoyl group, and reduction of the enoyl to an alkyl group, respectively (Figure 1B). A highly-reducing PKS has KR, DH, and ER domains whereas a partially-reducing PKS lacks either β-keto reductase (KR), dehydratase (DH), or enoyl reductase (ER) domain. The functions of individual domains have been dealt with in detail in other articles [23,24,25,28,29,30,31] and will not be elaborated upon in the current article. In this review, I mostly focus on non-reducing PKSs and the compounds they produce as most of the lichen metabolites connected to their genes are NRPKS-derived.

The NRPKSs have been classified into nine groups, corresponding to the supported monophyletic clades of the NRPKS phylogeny (Figure 1C). Each group codes for a specific category of compound, for instance, group I codes for orcinol derivatives, group II codes for melanins, group III codes for conidial yellow pigments, group IV codes for mycotoxins, such as aflatoxin/sterigmatocystin, and group V is implicated in the production of mycotoxins, such as desertorin, atrochrysone, etc. The lichen compounds linked to the genes belong to group I—orcinol derivatives, group VI—dibenzofurane derivatives (usnic acid), and group IX—ß-orcinol derivatives (Figure 1C). 

Each group contains several monophyletic clades, which have been named numerically, i.e., PKS1, PKS2, up to PKS48. Overall, nine groups contain about 50 PKSs. The PKSs associated with the synthesis of a lichen compound comprise PKS16 in group I, PKS23 in group IX, and PKS8 in group VI (Figure 1C). As more genomes are sequenced, novel supported monophyletic clades are being discovered. For instance, novel monophyletic clades were reported in the NRPKS phylogeny extrapolated by Singh et al. [15], suggesting that the number of PKSs in several groups exceeds the one known under the current classification. 

### 2.3. Biosynthetic Research on Non-Lichenized Versus Lichenized Fungi

Lichenized and non-lichenized fungi have similar biosynthetic potential—on average, about 24 and 35 biosynthetic gene clusters are present, respectively [32,33]. Despite this finding, the fungal secondary metabolites used in industry are predominantly of non-lichenized origin [34,35,36], and the biosynthetic potential of lichenized fungi mostly remains unexploited. For instance, fungal products are used, among other things, as immunosuppressive, hypocholesterolemic, antitumor, and antioxidant agents [34,37]. Some of the most common drugs of fungal origin constitute the cancer drug taxol, and the hypocholesterolemic drug lovastine [38]. Overall, alongside plant- and bacteria-derived drugs, fungal metabolites contribute significantly to medicines. In fact, of about 12,000 antibiotics that were known in 1955, approximately 22% were contributed by fungi [39,40]. 

While the cloning and characterization of PKS started about two decades ago for non-lichenized fungi, this field is only recently and slowly gaining ground for lichenized fungi. The first fungal PKS to be cloned was 6-methylsalicylic acid synthase (MSAS) from *Penicillium* in 1990 [41]. Later, the MSAS was also identified from *Aspergillus terreus* and then heterologously expressed in *A. nidulans* [26]. The first lichen PKS could be cloned only two decades later—the PKS16 from *Cladonia grayi* in 2011 [20]. The first lichenized fungal PKS was heterologously expressed in 2021 [42]. This lag in successful heterologous expression between non-lichenized and lichenized fungi is mostly because lichens are an obligate symbiosis from the fungal point of view [43,44], making it challenging to grow the lichen mycobiont effectively in an axenic culture and to obtain enough biomass within a reasonable amount of time in order to perform gene knock-out or knock-in experiments. Furthermore, the slow growth rate of lichens and the fact that the stimuli triggering secondary metabolite synthesis are either unknown or may not be reproducible in the laboratory further complicate NP-biosynthesis-related experiments. Even when the transcription experiments are successful in a heterologous host, the genes may not be translated and the molecule is thus not synthesized [20,21]. All these factors hindered the secondary metabolite-related experiments in lichens for quite some time. 

Genome-guided gene discovery has revolutionized biosynthetic research on lichenized fungi. This approach involves sequencing the metagenome of a lichen, followed by retrieving the fungal contigs via binning approaches and predicting biosynthetic genes using BGC prediction programs, such as antiSMASH [45,46]. For non-lichenized fungi, the genome-guided PKS discovery and linking of genes to molecules had already been accelerated around 2005 [1,47,48,49]. Several fungal genomes were sequenced, and their biosynthetic potential was revealed around this time and in the following years [50,51,52]. These studies showed that the biosynthetic capacity of fungi is far greater than the molecules produced by them, and indicating the presence of untapped biosynthetic potential in fungi. Similar studies on non-lichenized fungi, however, started only in the last few years, characterizing the biosynthetic potential of lichenized fungi [12,13,53,54,55]. These studies have shown that similar to non-lichenized fungi, lichenized fungi have a tremendous untapped/unexpressed biosynthetic potential. 

## 3. Part II: The Metagenomic Approach and Lichen Molecules Linked to Genes

### 3.1. Metagenomic-Guided Linking of Molecules to Genes

The metagenomic-guided linking of metabolites to genes is a boon for unconventional, difficult-to-culture organisms, such as lichens. The structures of most of the secondary metabolites of lichen are already known. This serves as a baseline for predicting the biosynthetic machinery and gene/s required for the synthesis of a metabolite. The BGC-identifying algorithms, such as antiSMASH [45,56], predict the clusters present in the genomes and classify them according to the core gene that is present in them as an NRPS cluster, PKS cluster, terpene cluster, etc. AntiSMASH also compares the BGCs to the MiBiG [57], the database of BGCs characterized from plants, bacteria, and fungi, to identify those BGCs that are most structurally and functionally similar. This step already narrows down the candidate gene/cluster search substantially. The estimated BGCs can be further grouped into sets of structurally similar BGCs by clustering programs, such as BiG-SCAPE [58] and BiG-FAM [59], to exclude the structurally and functionally divergent BGCs. The phylogenetic clustering of the genes provides further evidence of the function of PKS [24,42,46,60]. In most cases (at least for the studies linking lichen metabolites to genes), this narrows down the search to a single, most suitable candidate PKS.

Next, the presence of suitable accessory genes (e.g., oxidases, methyltransferases) further ascertains the function of the cluster and authenticates if the cluster is equipped to code for a particular metabolite. The presence of an expected accessory gene in the cluster, for instance, an OMT in an O-methylated product and a CYP450 in a molecule involving an oxidation step for its synthesis, brings further evidence that the cluster most probably codes for the molecule in question. 

### 3.2. Lichen Metabolites Linked to Genes

So far, the following lichen metabolites have been linked to genes: lecanoric acid (an orcinol didepside, Figure 2A), atranorin (a ß-orcinol depside, Figure 2B), grayanic acid (an orcinol didepsidone, Figure 2C), usnic acid (dibezofurane derivative, Figure 2D), gyrophoric acid (an orcinol tridepside), olivetoric acid (an orcinol didepside, Figure 2A), and physodic acid (an orcinol didepsidone, Figure 2C). Of these, the link has been experimentally validated for only two metabolites (Figure 3A,B), whereas, for the others, the inference has been based on genomic, phylogenetic and molecular data (Figure 3C–G).

#### 3.2.1. Grayanic Acid

Grayanic acid (C_21_H_22_O_7_) is an orcinol depsidone (Figure 2C and Figure 3F). It is a relatively rare metabolite reported only from *Neophyllis melacarpa* and *Cladonia grayi*. Grayanic acid PKS was the first lichen PKS to be linked to the biosynthesis of a secondary metabolite [20].

The grayanic acid cluster was identified from *Cladonia grayi* in 2011 by integrating genome mining, phylogenetics, and other molecular biology approaches (Figure 3F), i.e., degenerate polymerase chain reaction (PCR) and the expression data analyses of single-spore isolates induced to produce grayanic acid [20]. The candidate cluster was singled out by comparing mRNA and grayanic acid induction profiles and by comparing the accumulation of grayanic acid to the induction of PKS. The expression of only one PKS is correlated to grayanic acid accumulation. The corresponding cluster has a cytochrome P450 monooxygenase (CYP450) and an O-methyltransferase gene, which are essential for the synthesis of grayanic acid (Figure 2C). This further validated the candidate PKS as the most likely grayanic acid PKS.

#### 3.2.2. Atranorin

Atranorin (C_19_H_18_O_8_) is a ß-orcinol depside (Figure 2B and Figure 3B) with a broad-spectrum pharmacological property, e.g., analgesic, anti-inflammatory, antibacterial, antifungal, cytotoxic, antioxidant, and antiviral capabilities. This metabolite has been reported from several lichens, such as *Bulbothrix* spp., *Cladonia* spp., *Evernia prunastri*, *Hypogymnia, Imshaugia*, *Letharia, Parmelia*, *Physia* spp., *Pseudevernia furfuracea*, and *Umbilicaria* spp. Atranorin PKS has been so far identified from *Bacidea rubella* [12], *Pseudevernia furfuracea* [42,61]*, Cladonia rangiferina*, *Evernia prunastri*, *Letharia columbiana*, *Letharia lupina*, *Parmelia* spp., and *Stereocaulon alpinum* [42]. 

Atranorin is one of the two lichen secondary metabolites to be successfully expressed heterologously [42]. The corresponding PKSs were identified for the first time in 2020 by integrating genome mining, BGC clustering, and the phylogeny approach, followed by experimental confirmation via heterologous expression [42]. The authors generated a NRPKS phylogeny using seven taxa—six *Cladonia* spp. and *Stereocaulon alpinum*. They used gene network analysis to group BGCs into gene cluster families and found that only one family was shared by the two atranorin producers, *C. rangiferina* and *S. alpinum*. These PKSs formed an additional supported clade in the PKS phylogeny, which is specific to the ß-orcinol compounds. This discovery led to the addition of an novel group, group IX, to the NRPKS phylogeny, in order to accommodate PKS23, the atranorin PKS (Figure 1C). 

The authors then identified five other atranorin BGCs using the CORASON pipeline [58] on the OMT sequence from *C. rangiferina*. Four genes, i.e., atranorin PKS, CYP450, OMT, and a transporter, were syntenic among the atranorin-producing species: atr1 (PKS23), atr2, atr3, and atr4. These four genes are conserved among the atranorin producing taxa used in above-mentioned study. It is worth noting that the atranorin cluster from both *C. rangiferina* and *S. alpinum* contains 14 genes, but only three were sufficient to produce the compound in a heterologous system. 

The PKSs of group IX (Figure 1C; PKSs coding for methylated orsellinic acid derivatives) were considered possible candidates for atranorin synthesis. The corresponding cluster had two other tailoring enzymes required for atranorin synthesis, namely, a CYP450 and an O-methyltransferase (Figure 3B). The candidate PKS from *Stereocaulon alpinum*, along with CYP450 and O-methyltransferase, was heterologously expressed in the plant-pathogenic fungus *Ascochyta rabiei*. The knock-in strain expressing the three genes produced atranorin, thereby definitively linking the cluster to atranorin synthesis (Figure 3B).

The study by Kim et al. [42] is an important milestone in lichen biochemistry. Not only was it the first time that a lichen compound was produced by heterologous expression, it also involved expressing three different genes from a cluster. The authors used an unconventional host, a plant pathogenic fungus, instead of *Escherichia coli*, *Aspergillus nidulans*, or *A. oryzae*. It is noteworthy that previous attempts to heterologously express lichen PKS using conventional hosts had failed [21]. Although the genes were transcribed, translation was not successful and the target metabolite was not obtained in the knock-in host. Bertrand and Sorensen 2019 [21], for instance, inserted four PKSs from *Cladonia uncialis* into *A. oryzae* and found that all of them were transcribed but none translated. The successful transcription and translation of lichen PKSs and the requisite accessory genes in *Ascochyta rabiei* to heterologously express atranorin could pave the way forward for the validation of other lichen PKSs, a hurdle that has restricted lichen biosynthetic studies for many years. 

#### 3.2.3. Lecanoric Acid

Lecanoric acid (C_16_H_14_O_7_) is an orcinol didepside (Figure 2A and Figure 3A) reported from several lichens including, *Parmelia* sp., *Parmotrema* sp., *Umbilicaria* sp. as well as non-lichenized fungi as *Aspergillus* sp. etc. It is one of the simplest didepsides, with two phenolic rings, joined together by an ester bond; the phenolic rings lack any side chains. Lecanoric acid is the other lichen metabolite that is heterologously expressed and is directly linked to a secondary metabolite [62]. The host and the approach that was used in this study were, however, different from the one used by Kim et al. [42].

The authors identified a candidate NRPKS from the *Pseudevernia furfuracea* genome, published under the framework of another study [54]. They used the putative grayanic acid NRPKS as bait for this. The protein sequence was deduced from the candidate NRPKS from *P. furfuracea* and was reverse-translated to generate a DNA sequence with codons optimized for *Saccharomyces cerevisiae* to synthesize the intron-free gene (6345 bp). The synthetic, intron-free and codon-optimized PKS gene, driven by the *S. cerevisiae* glucose-regulated ADH2 promoter, was introduced into *S. cerevisiae* via an expression vector. Yeasts harboring an expression plasmid produced lecanoric acid, thereby directly linking this PKS to lecanoric acid synthesis.

This study by Kealey et al. [62], however, raises some questions: lecanoric acid (Figure 3A) is not produced by *P. furfuracea* in nature; instead, the orcinol compounds reported from *P. furfuracea* are olivetoric acid (a didepside, Figure 3C) and physodic acid (a didepsidone, Figure 3E). As there is only one PKS16 homolog (depside PKS) present in *P. furfuracea*, the “lecanoric acid” PKS is, thus, involved in the synthesis of olivetoric/physodic acid. The experimental justification for “how a putative olivetoric/physodic acid PKS produces lecanoric acid in yeast” is yet derived but it was proposed that this could be attributed to the lack of appropriate starter units in yeast [61]. This conundrum gave rise to an interesting hypothesis regarding polyketide synthesis, i.e., the starter unit is crucial to enhancing the diversity of secondary metabolites, as the same PKS may produce different compounds in the presence of different starter units [61]. In fact, the authors themselves tried to feed different starter units but they always obtained lecanoric acid. 

The study by Kealey et al. [62] marks an important milestone in lichen biochemical studies, first because it unequivocally links an NRPKS, PKS16, to the synthesis of an orcinol depside (Figure 2A), confirming the role of these PKSs in the synthesis of orcinol compounds, and second, because it highlights the importance of the host and the starter units in the heterologous expression of lichen secondary metabolites.

### 3.3. Molecules That Are Bioinformatically Linked to Their Respective Genes/Clusters 

#### 3.3.1. Usnic Acid 

Usnic acid (C_18_H_16_O_7_) is a dibenzofuran derivative (Figure 2D and Figure 3G). The usnic acid gene cluster is one of the best-studied lichen biosynthetic clusters, both genomically and experimentally, as well as across species. It is reported from several lichenized fungal genera, including *Alectoria*, *Cladonia*, *Evernia*, *Lecanora*, *Ramalina*, *Usnea*, and *Usnochroma*. The first most-likely gene cluster for usnic acid was identified from *Cladonia uncialis* in 2016 [55]. To date, this cluster has been identified from about 20 lichenized fungi, including *C. rangiferina, C. metacorallifera*, *Lobaria pulmonaria*, *Nephromopsis pallescens* [63], and *Usnea* spp. (Figure 3G). The putative usnic acid PKS belongs to group VI (Figure 1C).

The first usnic acid gene cluster (from *Cladonia uncialis*) was identified via genome mining approach [55]. The authors narrowed down the most suitable candidate NRPKS from *C. uncialis* based on the biosynthetic requirements for usnic acid synthesis, i.e., a, NRPKS with a carbon methylation (cMT) domain, a terminal Claisen cyclase (CLC) domain, and an oxidative enzyme (mostly CYP450) for the dimerization of methylphloracetophenone to usnic acid. Only one candidate was found to fulfill the above-stated requirements and the corresponding genes were also transcriptionally active, suggesting that this may be the most likely usnic acid cluster. 

Later, a phylogeny of NRPKSs from 46 lichenized fungi, including *Cladonia uncialis*, showed the presence of putative usnic acid PKS in all the producers and their absence in the non-producers [13]. Although the overall cluster composition varies, all producers have the usnic acid PKS and a CYP450 gene, as found in *C. uncialis*, building further evidence that the candidate usnic acid PKS that has been identified in *C. uncialis* was the one involved in usnic acid synthesis. It is to be noted that in all these taxa, usnic acid was a major metabolite and was identified using TLC or HPLC. Recently, usnic acid (and the corresponding cluster) was reported from *C. rangiferina*, formerly considered a non-producer, using LC-MS, a more sensitive detection technique [14]. This indicates that the presence of usnic acid might be more abundant in lichenized fungi than we currently know. Implementing more sensitive metabolite detection techniques will be essential to understand the taxonomic breadth and evolution of this gene cluster. 

#### 3.3.2. Olivetoric/Physodic Cluster

Olivetoric acid (C_26_H_32_O_8_) and physodic acid (C_26_H_30_O_8_) (Figure 3C,E) are potent cytotoxic, antimicrobial, and anti-oxidative agents. They are orcinol derivatives and comprise isostructural depside-depsidone pairs, i.e., they are structurally similar compounds except for the fact that olivetoric acid is an orcinol didepside (Figure 2A), whereas physodic acid is the an orcinol didepsidone, most probably derived from the oxidation of olivetoric acid by a CYP450 (Figure 2C). These two are reported to co-occur in the same species, often within the same sample but in different proportions—*Pseudevernia furfuracea*, *Cetraria ciliaris*, *Ramalina leoidea*, and *Hypogymnia lugubris* [61,64,65,66,67] The corresponding organisms are considered to be chemical races, based on the compounds present in them, but, often, both metabolites are reported to be present in the same sample. 

The olivetoric/physodic cluster was recently identified from *Pseudevenia furfuracea*, based on the metagenomic and transcriptomic data [61]. The chemical race of the sample was first determined by HPLC, followed by the whole-genome sequencing of the races. Comparative genomics revealed only one PKS16 in both samples. The PKS16 from the two chemotypes was homologous, as expected, since the backbone molecule for both molecules is the same, except for the fact that there is an additional ether bond in physodic acid. Given that PKS16 catalyzes only ester bond synthesis and, therefore, depside formation (Figure 2A), it is quite likely that PKS produces the depside, olivetoric acid, in both chemotypes but in the physodic acid chemotype the CYP450 is active and catalyzes the synthesis of the depsidone.

The study by Singh et al. [61] shows that a cluster may be involved in the synthesis of more than one compound. This has been shown in non-lichenized fungi, but this was the first indication of the occurrence of such a phenomenon in lichenized fungi. In addition, the same PKS, when expressed heterologously in yeast, produced lecanoric acid, which is, again, a didepside, further demonstrating the promiscuity of this PKS. The same PKS synthesizes three different molecules, depending upon the starter unit (acetyl CoA, malonyl CoA, hexanoyl CoA, or octanoyl CoA) and the accessory enzymes that are activated (CYP450 is present upstream of the PKS when activated; it catalyzes the ether bond formation, leading to physodic acid synthesis).

#### 3.3.3. Gyrophoric Cluster

Gyrophoric acid (C_24_H_20_O_10_) is a tridepside (Figure 3D) that is synthesized by several *Umbilicaria* species, *Cryptothecia rubrocincta*, *Lecidea fuscoatra*, *Montanelia tominii*, *Parmotrema tinctorum*, *Punctelia borreri* and *Xanthoparmelia pokomyi* etc. Usually, it is accompanied by several other structurally related molecules, such as umbilicaric acid, hiascic acid, and lecanoric compounds.

The most-likely gyrophoric acid cluster has been recently identified from nine *Umbilicaria* species [15]. Interestingly, although the species had different minor metabolites (*U. deusta* and *U. grisea* had umbilicaric acid, while *U. sprodochroa* had umbilicaric and lecanoric acid), only one copy of PKS16 was present in all *Umbilicaria* species included in the study by Singh et al. [15]. The corresponding PKS, PKS16, was highly homologous among all species. This study further indicates that PKS16 may be promiscuous and may synthesize a few structurally-related compounds. Another highlight of this study was that it was the first work identifying a tridepside PKS. This study showed that the domain structures of a tridepside PKS and a didepside PKS are the same, i.e., a KS-AT-ACP-ACP-TE, and that a third ACP domain might not be required for tridepside synthesis.

## 4. What We Have Learned So Far: Critical Takeaways and Novel Emerging Concepts 

### 4.1. Metagenomic and Molecular Predictions Mostly Nail Down the Target Gene Precisely

To heterologously express the lecanoric acid and atranorin PKS, the authors first selected the most-likely candidates based on the genome mining of biosynthetic genes and the phylogenetic grouping of the PKSs. In both cases, the authors ended up with one candidate that fitted the biochemical criteria for producing the corresponding molecule. This is also the case for other molecules linked to their corresponding genes via genome mining, i.e., only one candidate PKS per species fits the criteria to produce a molecule, e.g., gyrophoric acid [15], olivetoric acid [61], atranorin [12,61], and usnic acid [13]. This suggests that, in most cases, one species harbors only one copy of PKS16, PKS23, or usnic acid PKS and the chemistry of the compound combined with the information on the required biosynthetic machinery in most cases enables the precise prediction of the most-likely candidate.

### 4.2. The Biosynthetic Potential of an Organism Far Exceeds the Compounds Produced/Detected

This phenomenon has been demonstrated by several studies; therefore, it will only be dealt with briefly here. Genomic studies on lichens show that lichens harbor, on average, 20–70 BGCs but there are usually fewer than 10 metabolites known from a species. Moreover, a lichen mycobiont usually has different classes of BGCs—NRPS, reducing PKS, NRPKSs, terpenes, hybrid PKS-NRPS, and RiPP—but the compounds known from lichens are mostly NRPKS-derived, e.g., depsides and depsidones, melanins, anthraquinones, dibenzofurane derivatives, etc. This suggests that the realized metabolic potential of lichens is only a fraction of their actual potential. However, using more sensitive detection techniques unravels several previously undetected metabolites, indicating that the gap between actual and realized metabolic potential might simply be an outcome of the low sensitivity of detection techniques. 

For instance, initially, thin-layer chromatography (TLC), a technique with a detection limit of ~500 ng/mL, was the standard method used to detect lichen metabolites. This was then taken over by advanced techniques, such as HPLC, which are more sensitive (1 μg to 1 pg/mL) depending upon the type of HPLC being used [68,69]. More recently, more sensitive methods, such as mass spectrophotometry (MS), have been used, which lowered the detection limit to the level of ng/mL (LC-MS) and have the potential to detect about 200 metabolites over a short running time [70,71]. Recently, ultra-performance liquid chromatography (UPLC) and UPLC-tandem mass spectrometers (UPLC-MS/MS) have also been implemented and have led to the detection of previously unknown compounds from various species [70,72,73]. For instance, *Cladonia rangiferina* was thought to be incapable of producing usnic acid (based on TLC), until the extracts were examined by MS [14]. Similarly, several previously undetected compounds were reported from whole thallus extracts of *Evernia prunastri*, *Hypogymnia physodes*, and *Ophioparma ventosa*, when examined via MS [73]. As more sophisticated detection techniques are becoming common, this gap is expected to narrow down at a relatively rapid rate. Nonetheless, it is unlikely that the number of metabolites will match that of biosynthetic genes - it is likely that several silent or orphan clusters are present in a genome, as happens in bacteria and non-lichenized fungi.

### 4.3. One Cluster May Code for Several Structurally Related Compounds

One of the most surprising aspects of the metagenomic studies on the secondary metabolites and PKS genes of lichen is that they reinforced the idea that instead of one dedicated PKS for each molecule, one PKS may code for a few structurally similar molecules. The final product depends on the starter unit, the heterologous host used (see Section 3.2.3), external triggers, and the set of enzymes expressed along with the PKS. For instance, the same PKS codes for lecanoric acid (Figure 3A), olivetoric acid (Figure 3C), and physodic acid (Figure 3E) [54,61,62]. Another such example is *Umbilicaria* PKS16. Genome mining, combined with phylogenetic clustering, suggests that the same PKS codes for umbilicaric, gyrophoric, lecanoric, and hiascic acid [15]. Interestingly, these compounds often show very different bioactivity. Lecanoric acid, for instance, has antifungal and anti-proliferative properties, whereas physodic and olivetoric acid have cytotoxic and anti-oxidative effects [9]. Likewise, olivetoric acid displays antifungal properties, whereas physodic acid does not [74]. The production of different compounds by the same PKS may be temporally and seasonally separated or concomitant. 

One cluster coding for several molecules provides an excellent baseline for combinatorial biosynthesis, which involves using genetic engineering to modify biosynthetic pathways for producing new metabolites. Thus, the “flexibility” of PKS toward the starter unit further expands the pharmaceutical potential of lichens.

### 4.4. The Same PKS May Produce Different Compounds in the Presence of Different Starter Units

Starter-unit selection, as performed by the SAT domain of the PKS core gene, is a key step in the biosynthesis of polyketide natural products and is an important contributor to the diversity of the compounds produced by a PKS [25,75,76]. These domains accept acetyl-CoA, either directly or as a more complex starter unit, such as butanoyl CoA, hexanoyl CoA, or octanoyl CoA, which is transported to the PKS by a specialized fatty acid synthase (FAS) or a highly reducing PKS. The SAT domain has been shown to be highly selective for a particular starter unit and accepts that unit preferentially. For instance, although the SAT domain (of the PKS grouping in the norsolorinic acid synthase clade) from *Coccidioides immitis* accepts different starter units, i.e., malonyl, acetyl, butanoyl, henoyl, and octanoyl-CoA, it displays a 4-fold preference for octanoyl-CoA, compared to hexanoyl-CoA [23]. The same PKS produces different metabolites in the presence of hexanoyl and octanoyl-CoA. 

The lack of a “natural” starter unit in a heterologous host could result in a different metabolite than that known from the organisms from which the PKSs originate. For instance, Kealey et al. [62] heterologously expressed *Pseudevernia furfuracea* orcinol-depside-PKS, PKS16, in *E. coli,* and thereby obtained lecanoric acid instead of olivetoric acid. This is remarkable as lecanoric acid has never been reported from *Pseudevernia furfuracea* in nature. This anomaly was hypothesized to have been caused by the absence of an appropriate starter unit in the heterologous host used (*E. coli*) [61]. The flexibility of the PKS’s SAT domain toward starter units presents an opportunity for combinatorial biosynthesis to engineer different metabolites from the same PKS using different starter units. 

### 4.5. KS, PT, or Full-Length NRPKS Generate Congruent Phylogenies, Even Though the Alignment May Be Cleaner When Only the Highly Conserved Domains, Such as KT and PT, Are Used

Fungal NRPKS phylogenies have been inferred using KS (ketoacyl synthase domain) [60] and PT (product template domain) [24,77] domains, owing to their highly conserved nature, or using the entire NRPKS gene [12,15,24]. The first phylogeny of fungal PKSs, published in 2002, was based on the KS domain genealogy [60]. Later, a few studies inferred NRPKS phylogenies based on a single conserved domain, as well as derived from full-length NRPKS genes, and found them to be highly congruent [24,78]. For instance, Pizarro et al. [13] investigated the presence of the usnic acid cluster in *Lecanoromycetes* based on the KS domain, as well as using the full usnic acid PKSs. Similarly, Liu et al. [24] inferred the PKS phylogeny of 187 strains of ascomycetes, based on both, the PT domain and full-length NRPKS sequences, and found the phylogenies to be congruent. Some recent studies, based only on full-length NRPKSs, obtained well-supported groups corresponding to Figure 1C. For example, Gerasimova et al. [12], Singh et al. [61], and Singh et al. [15] generated NRPKS phylogenies to infer the atranorin (Figure 3B), olivetoric acid (Figure 3C), and gyrophoric acid clusters (Figure 3D), respectively. These studies suggest that the PKS phylogeny, inferred from both, a single conserved domain or a full-length NRPKS, provide a phylogenetic tree with monophyletic clades corresponding to the nine PKS groups (Figure 1C).

### 4.6. Not All Genes of a Cluster Are Involved in Metabolite Synthesis but May Be Involved in Its Transportation or Regulation

Several studies have shown that only a few genes of the cluster are essential to synthesize the metabolite heterologously. In case of depsides without side-chain modifications, such as lecanoric acid (Figure 3A), PKS alone is sufficient (Figure 2A). This has been proven by the heterologous expression of the *Pseudevernia furfuracea* PKS16. On the other hand, for atranorin and usnic acid, apart from the PKS, other genes are required to produce the compound (Figure 2B and Figure 2D respectively). In general, not all the genes of a biosynthetic cluster, despite being conserved across species, are required to produce the compound. For instance, for atranorin synthesis, the expression of three genes, i.e., atranorin PKS, CYP450, and OMT, was sufficient to produce the compound in the heterologous system, although the cluster contains about 10–14 genes, including some highly conserved ones. Several reasons could be responsible for this phenomenon. Some genes of the cluster may have a regulatory or transport-related function and, therefore, are not required for the synthesis of the compound. Another reason could be that some genes of the cluster may be under the control of pleiotropic (controlling the production of multiple antibiotics and/or morphological development) and global regulators (controlling central metabolic genes and pleiotropic regulatory genes, as well as cluster-situated regulators), located outside the BGC [16,79]. Therefore, these genes are activated only under specific conditions, when triggered by external regulators. 

Another thing to consider would be that the cluster boundaries are predicted by an algorithm and the number of genes that are included in a cluster may not necessarily indicate that they are all part of the same cluster. It is quite likely that the program may not be precise in predicting the cluster boundaries and may underestimate or overestimate the cluster size.

## Figures and Tables

**Figure 1 jof-09-00160-f001:**
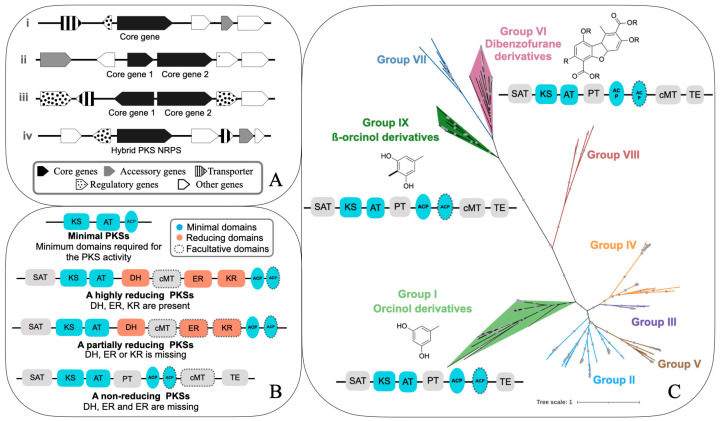
Biosynthetic gene clusters, types of PKSs (based on domains), and the evolutionary relationships of non-reducing PKSs. (**A**) Different kinds of biosynthetic gene clusters, depending upon the core genes: (i) a cluster containing a single core gene—this could be a type-I NRPKS (usually 5000–8000 bp) or a type-III NRPKS (~800–1300 bp), an NRPS (~3000–4000 bp), terpene (~800 bp), indol (800–1300 bp), or ribosomally synthesized and post-translationally modified peptides (RiPP, 600–900 bp) BGC; (ii) a cluster containing two core genes—this could be an NRPKS and a reducing PKS, or both reducing PKSs, or both NRPKSs; (iii) a BGC with two core genes in the opposite orientation; (iv) a hybrid BGC with both an NRPS and an NRPKS. (**B**) Domain composition of minimal PKSs, highly-reducing PKSs, partially-reducing PKSs, and non-reducing PKSs. Dotted borders denote that the domain is facultative. (**C**) A representative 1000-bootstrap maximum-likelihood NRPKS tree, showing the nine NRPKS groups. Colored clades represent the groups containing the PKSs, linked to a molecule. The structure of the molecules coded by these clades and the domains of the corresponding PKS are shown in the figure (R = C_n_H_n_). (adapted from Singh et al. 2021 [15]; dataset downloaded from supplemental material Table S2, for details on the dataset, alignment and tree annotation please refer to Singh et al. 2021 [15]). The final data set consisted of amino acid sequences of 229 NR-PKSs from 18 species belonging to five LFF genera-*Dermatocarpon*, *Cladonia*, *Pseudevernia*, *Stereocaulon*, and *Umbilicaria*.

**Figure 2 jof-09-00160-f002:**
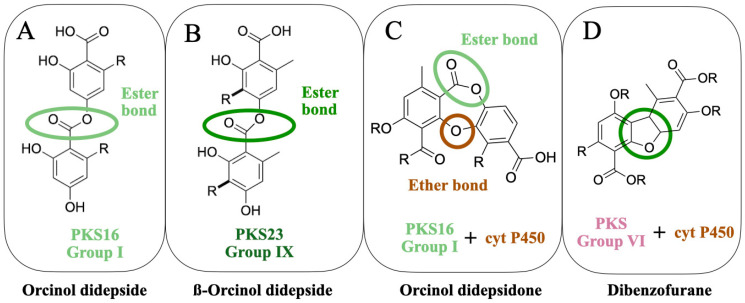
The molecular structures of four classes of lichen compounds linked to the genes. The colored circles denote the characteristic bond of a compound-ester bond present in a depside (**A**), the ester bond of a ß-orcinol depside (**B**), the ester and ether bond present in a depsidone (**C**), and the characteristic dibenzofurane bond (**D**).

**Figure 3 jof-09-00160-f003:**
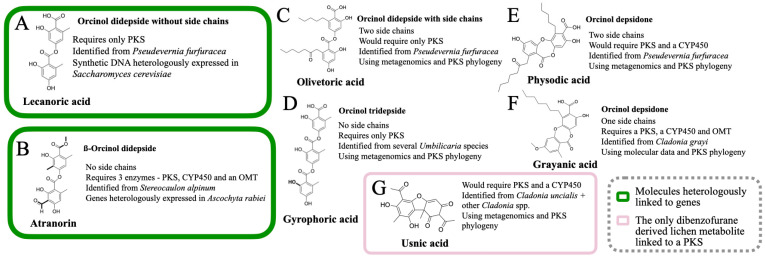
The lichen compounds that are linked to a PKS. The compounds highlighted in green are the ones in which the link between the molecule and the PKS was also verified by heterologous expression. (**A**) lecanoric acid, (**B**) atranorin, (**C**) olivetoric acid (**D**) gyrophoric acid, (**E**) physodic acid, (**F**) grayanic acid and (**G**) usnic acid.

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
