# Peer review of "Linking Lichen Metabolites to Genes: Emerging Concepts and Lessons from Molecular Biology and Metagenomics"

_jof, 2023, doi:10.3390/jof9020160_

Round 1

Reviewer 1 Report

In this manuscript, the author reviews on the lichen metabolites linked to their corresponding biosynthetic genes and highlights the application of metagenomics-based strategies to establish this link as well as summarizes the direct and serendipitous lessons learned from those studies. The manuscript is well-organized, and the cited references are appropriate and adequate. The paper will be of interest to researchers working on biologically active natural products and their synthesis/biosynthesis. Therefore, the paper merit publication in the Journal of Fungi after minor revisions.

Major comments:

1. This review is focused on lichen metabolites and its corresponding biosynthetic gene clusters. It may be more informative to readers to include “lichen metabolites” in the keyword, which is too general in its current form “secondary metabolites”.

2. The presentation quality will be further improved by additional proofreading and confirmation of appropriate terminology used throughout the paper. As a few examples: page 9, lines 394-397, 398-399.

3. The references need to be edited much more carefully, especially for the journal’s name, for example, references 25, 30, 60 as well as doi numbers, references 32, 51, 52.

Minor points

4. Line 18, “these studies in not available” do you mean “these studies are not available”

5. Figure 1C, the second chemical structure in group IX should be revised. The OH group is far away from its core structure.

6. Several strain names should be italic type, such as Line 152, Penicillium; line 254, S. alpinum; lines 285, 287, S. cerevisiae; lines 290, 291, P. furfuracea, and so on.

7. Line 218, the numbers of C, H, O should be subscript.

8. Figure 3B, “cut P450” should be “cyt P450”.

9. Line 331, delete “of”

10. Line 472, define the full name of “PT”.

Author Response

Point by point response to reviewer 1

1. This review is focused on lichen metabolites and its corresponding biosynthetic gene clusters. It may be more informative to readers to include “lichen metabolites” in the keyword, which is too general in its current form “secondary metabolites”.

Done

  1. The presentation quality will be further improved by additional proofreading and confirmation of appropriate terminology used throughout the paper. As a few examples: page 9, lines 394-397, 398-399.

Done. The terminology has been checked throughout the manuscript

  1. The references need to be edited much more carefully, especially for the journal’s name, for example, references 25, 30, 60 as well as doi numbers, references 32, 51, 52.

Done

  1. Line 18, “these studies in not available” do you mean “these studies are not available”

Lines 16-19. We rephrased the sentence as follows: Although the structures of most of the lichen secondary metabolites are well-documented, a comprehensive review on the metabolites linked to their genes, strategies implemented to establish this link and crucial takeaways from these studies is not available.

  1. Figure 1C, the second chemical structure in group IX should be revised. The OH group is far away from its core structure.

Thank you very much for pointing this out. We revised the figure to rectify the suggested issue.

  1. Several strain names should be italic type, such as Line 152, Penicillium; line 254, S. alpinum; lines 285, 287, S. cerevisiae; lines 290, 291, P. furfuracea, and so on.

Done. The species name have been italicized everywhere.

  1. Line 218, the numbers of C, H, O should be subscript.

Done

  1. Figure 3B, “cut P450” should be “cyt P450”.

Thank you very much for pointing this out. We have revised the figure to rectify the this issue.

  1. Line 331, delete “of”

Done

  1. Line 472, define the full name of “PT”.

Done

Reviewer 2 Report

Dear Editor, dear Author, 

this is a very well-written, concise, clear and comprehensive review about the knowledge of lichen metabolites and genes encoding them. 

I fully support the publication of this work, which requires only minor changes that I have suggested in the revised pdf file attached here.

I strongly recommend that the genus and species scientific names are carefully check throughout text and References and must be written in cursive/italics. 

I do not agree with the style of the journal to write all nouns with capital letters in the references, as this is grammatically incorrect for the English language, but it is used only in German. This type of format leads to the incorrect writing of species names, for which only the genus name goes with capital letter and the species epitheton must not have capitals. I hope in your understanding. 

Author Response

Dear Reviewer,

Thank you very much for your comments of my manuscript.

I appreciate the thorough and constructive suggestions. I have addressed all your comments and provided point-by-point responses to all concerns raised by them. The line numbers refer to the numbers in the “manuscript-revised-changes-accepted” file. The suggestions mentioned in the pdf of the manuscript have been directly included in the revised version of the manuscript.

Sincerely,

Garima

Major comments

check for way of citing figure in text, so far and below writen in different ways:  Fig 1A, Fig1B, Fig. 1B, Fig. 1C, Fig 1C

Done. This has been synchronized throughout the manuscript

Figure legend 2 I suggest to change this into: Molecular structures of four classes of lichen compounds.

Done

Reviewer 3 Report

The molecular genetics behind the production of lichen secondary metabolites is a very interesting and important topic among recently studied and less known topics in lichenology.

The review is well structured consisting of 3 main chapters, analyzing non-reducing polyketide synthases comparing the biosynthetic research on non-lichenized versus lichenized fungi; the lichen metabolites that have been connected to genes; then summarizing the novel concepts of secondary metabolite synthesis and biosynthetic genes function – as a take-away messsage.

The review considered 80 publications selecting well the most important and valuable ones available in this field.

The logical structure of the review is coupled with the clear style of the author. It makes the paper of a difficult topic an easy to follow reading.

Only a few small mitakes/misspellings were found listed below.

Please, check carefully if all abbreviations are given, e.g. PP in line 96, GRA in line 225

Please, improve these mostly formal mistakes.

line 5

add affiliation

line 18

„in” to be replaced with „is”

line 107

change „nrPKS” to „NRPKS”

line 112

replace „auch” with „such"

line 125

change „genes” to „gene”

line 156

change „being lichens” to „lichens being”

line 207

„an” to „a” ß-………

line 218

lower the Fonts of figures in „C21H22O7” to „C21H22O7

line 267

change „E. coli” to „Escherichia coli”, „A. nidulans” to „Aspergillus nidulans”

line 270

change „C. uncialis” to „Cladonia uncialis”

line 271

change „Aspergillus oryzae” to „A. oryzae”

line 285

change „S. cerevisiae” in italics „Saccharomyces cerevisiae”

line 287

„S. cerevisiae” italics

lines 290, 291

„P. furfuracea” italics

line 294

change „How come then” to „How does it happen that”

line 331

the first part of the sentence („Later, a of phylogeny”) is not clear – please, reword

line 373

„This” should be replaced with „Gyrophoric acid

line 436

„Umbilicaria” – italics; „combine” to „combined”

line 456

„Coccidioides immitis” - italics

line 461

„know” to „known”

line 462

insert after „Kealey et al.”: „[62]”

line 463

a space is missing in „olivetoricacid”, change to „olivetoric acid”

line 499

change „reasons for responsible” to „reasons responsible”

References

42 - authors are missing:

Kim W, Liu R, Woo S, Kang KB, Park H, Yu YH, Ha H-H, Oh S-Y, Yang JH, Kim H, Yun S-H, Hur J-S. 2021. Linking a gene cluster to atranorin, a major cortical substance of lichens, through genetic dereplication and heterologous expression. mBio 12:e01111-21. https://doi.org/10.1128/mBio.01111-21.

Author Response

Dear Reviewer,

Thank you very much for your comments of my manuscript.

I appreciate the thorough and constructive suggestions. I have addressed all your comments and provided point-by-point responses to all concerns raised by them. The line numbers refer to the numbers in the “manuscript-revised-changes-accepted” file.

Sincerely,

Garima

RESPONSE TO THE COMMENTS ON THE MANUSCRIPT

  • Please, check carefully if all abbreviations are given, e.g. PP in line 96, GRA in line 225

We have changed GRA to grayanic acid everywhere in the manuscript to avoid confusion.

Please, improve these mostly formal mistakes.

  • line 5 add affiliation

Done. Line 5

  • line 18 „in” to be replaced with „is”

Done

  • line 107 change „nrPKS” to „NRPKS”

Done

  • line 112 replace „auch” with „such"

Done

  • line 125 change „genes” to „gene”

Done

  • line 156 change „being lichens” to „lichens being”

Done. Line 157. The sentence now reads: This lag in the successful heterologous expression between the non-lichenized and li-chenized fungi is mostly because, lichens being an obligate symbiosis from the fungal point-of-view

  • line 207 „an” to „a” ß-………

Done

  • line 218 lower the Fonts of figures in „C21H22O7” to „C21H22O7

Done.

  • line 267 change „E. coli” to „Escherichia coli”, „A. nidulans” to „Aspergillus nidulans”

Done

  • line 270 change „C. uncialis” to „Cladonia uncialis”

Done

  • line 271 change „Aspergillus oryzae” to „A. oryzae”

Done

  • line 285 change „S. cerevisiae” in italics „Saccharomyces cerevisiae”

Done

  • line 287 „S. cerevisiae” italics

Done

  • lines 290, 291„P. furfuracea” italics

Done

  • line 294 change „How come then” to „How does it happen that”

Done. Lines 295-296. The text now reads: How does it happen that a putative olivetoric/physodic acid PKS produces lecanoric acid in yeast?

  • line 331 the first part of the sentence („Later, a of phylogeny”) is not clear – please, reword

Done. Lines 332-334. The sentence was modifies as follows: Later, a phylogeny of NRPKSs from 46 lichenized fungi including Cladonia uncialis showed the presence of putative usnic acid PKS in all the producers and their absence in the non-producers.

  • line 373 „This” should be replaced with „Gyrophoric acid

Done. Line 219 Grayanic acid (C21H22O7) is an orcinol depsidone.

  • line 436 „Umbilicaria” – italics; „combine” to „combined”

Done

  • line 456 „Coccidioides immitis” - italics

Done

  • line 461 „know” to „known”

Done

  • line 462 insert after „Kealey et al.”: „[62]”

Done

  • line 463 a space is missing in „olivetoricacid”, change to „olivetoric acid”

Done

  • line 499 change „reasons for responsible” to „reasons responsible”

Done

  • References, 42 - authors are missing:

Kim W, Liu R, Woo S, Kang KB, Park H, Yu YH, Ha H-H, Oh S-Y, Yang JH, Kim H, Yun S-H, Hur J-S. 2021. Linking a gene cluster to atranorin, a major cortical substance of lichens, through genetic dereplication and heterologous expression. mBio 12:e01111-21. https://doi.org/10.1128/mBio.01111-21.

Done. The reference was formatted to include the missing information
